# Correlation between Metabolic Rate and Salinity Tolerance and Metabolic Response to Salinity in Grass Carp (*Ctenopharyngodon idella*)

**DOI:** 10.3390/ani11123445

**Published:** 2021-12-03

**Authors:** Pathe Karim Djiba, Jianghui Zhang, Yuan Xu, Pan Zhang, Jing Zhou, Yan Zhang, Yiping Luo

**Affiliations:** 1Key Laboratory of Freshwater Fish Reproduction and Development (Ministry of Education), School of Life Sciences, Southwest University, Chongqing 400715, China; pathekarim@email.swu.edu.cn (P.K.D.); zzjh81@email.swu.edu.cn (J.Z.); xuyuan224@email.swu.edu.cn (Y.X.); z461215098@email.swu.edu.cn (P.Z.); 2Clinical School, Chongqing Medical and Pharmaceutical College, Chongqing 401331, China; zj4321228@sina.com (J.Z.); seaheart1@163.com (Y.Z.)

**Keywords:** *Ctenopharyngodon idella*, gills, maximum metabolic rate, resting metabolic rate, salinity tolerance

## Abstract

**Simple Summary:**

The association between the metabolic rate and salinity tolerance in stenohaline freshwater fish could affect how fish adapt to changes in environmental salinity. In Experiment I, the metabolic rates and upper salinity tolerance limit of the grass carp were determined individually, and we aimed to test whether an association existed between the salinity tolerance capacity and both the resting metabolic rate and maximum metabolic rate. In Experiment II, the effects of increasing salinity on metabolic rates, gill histology, and Na^+^-K^+^-ATPase activities were determined in grass carp. The results suggest that a lower metabolic rate may not necessarily allow for a better salinity tolerance capacity of grass carp. Salinity-induced changes in the gill surface contribute more to ion exchange capacity than to oxygen uptake capacity.

**Abstract:**

The metabolic rate could be one of the factors affecting the salinity tolerance capacity of fish. Experiment I tested whether metabolic rates correlate with the upper salinity tolerance limit among individual grass carp by daily increasing salinity (1 g kg^−1^ day^−1^). The feeding dropped sharply as the salinity reached 10 g kg^−1^ and ceased when salinities exceeded 11 g kg^−1^. The ventilation frequency decreased weakly as salinity increased from 0 to 12 g kg^−1^ and then increased rapidly as salinity reached 14 g kg^−1^. The fish survived at salinities lower than 14 g kg^−1^, and all fish died when salinity reached 17 g kg^−1^. The upper salinity tolerance limit was not correlated with metabolic rates. Therefore, a lower metabolic rate may not necessarily allow for better salinity tolerance capacity. Experiment II tested how different salinities (0, 0.375, 0.75, 1.5, 3, and 6 g kg^−1^ for 2 weeks) affect the metabolic parameters of grass carp. The changes in the resting metabolic rate with increasing salinity could be explained by the relative changes in interlamellar cell mass and protruding lamellae. The maximum metabolic rate remained constant, suggesting that the salinity-induced changes in the gill surface had a minor effect on oxygen uptake capacity.

## 1. Introduction

The low-level increase in salinity is one of the environmental changes that freshwater fishes are experiencing in estuaries and aquaculture farms located along coasts due to drought, evaporation, and seawater intrusion induced by climate change [1,2,3]. Increases in salinity change the osmotic movement of water and the diffusion of ions between the water and fish body and affect the structural, biochemical, physiological, and life history processes of fish [4,5,6,7,8,9,10]. Teleost fishes maintain internal osmolalities of approximately 300–400 mosmol kg^−1^ (isosmotic to 9–12 g kg^−1^ salinity) [11,12,13,14,15]. As salinity increases below the isotonic level, the difference in salt concentrations between the water and plasma of fish decreases, and the rates of both salt loss and water gain drop (plasma hyper-osmoregulation). As salinity increases above the isotonic level, euryhaline fishes switch to regulation by a completely different set of osmoregulation mechanisms (plasma hypo-osmoregulation). However, stenohaline fishes do not possess the mechanisms to perform plasma hypo-osmoregulation and have lower tolerability to high salinity than euryhaline fishes.

It has been proposed that the tolerance capacity of fish to environmental stress may be closely related to metabolic performance variables, such as the resting metabolic rate (the minimum energy required to maintain basal demands) and maximum metabolic rate (the highest rate of aerobic metabolism) [16,17,18]. However, a compromise may exist between gaseous and ionic exchange across the gills of fish [19,20]. A higher resting metabolic rate may correspond to fast ventilation flow through the gills [16,21,22], while a higher maximum metabolic rate may represent a larger gill surface area [23]. It can be hypothesized that both of these conditions may positively affect ion exchange between the water and fish bodies and then result in a high vulnerability to salinity higher than isotonic conditions. Therefore, these metabolic variables may play negative roles in the salinity tolerance capacity of fish, resulting in individuals with both higher resting and maximum metabolic rates having lower salinity tolerance capacities.

Freshwater stenohaline fishes live in hypotonic environments, and they have to deal with diffusive ion losses by transporting salts from water, and osmotic water gain by excreting dilute urine [11,15]. The uptake of ions in freshwater fish is an energy-expensive process and contributes significantly to the resting metabolic rate of fish [11,24]. An increase in salinity below the isotonic level may reduce the osmotic gradient between the water and fish body and can be hypothesized to reduce the energy cost of osmoregulation of freshwater fish. Therefore, it can be assumed that the resting metabolic rate of freshwater fish would decrease as the salinity level increases below the isotonic level. However, the response of the resting metabolic rate to salinity varies depending on fish species, life habitats, and salinity levels [9,25,26]. Although the resting metabolic rate of some fish species was lower under near-isosmotic salinity than under hypo-osmotic salinity [24,27,28,29,30], the resting metabolic rate of other fish species showed no changes, linear increases, and bell-shaped changes with increasing salinity [9,14,19,25,31,32,33]. In addition, the lamellar morphology of the fish gills may change under varying salinity to regulate osmotic water movement and ion diffusion [14,24,32,33], which could be hypothesized to affect the respiratory gas exchange capacity of fish. Since the maximum metabolic rate is limited by the respiratory gas exchange capacity of fish [23], an increase in salinity would affect the maximum metabolic rate of fish. However, the available information on the effects of salinity on the maximum metabolic rate of stenohaline freshwater fish is limited.

The grass carp (*Ctenopharyngodon idella*) is a species of freshwater fish native to lakes, ponds, and large rivers in eastern Asia [34]. This fish has been introduced in many countries, mainly for aquaculture and aquatic macrophyte control, and is present almost everywhere in the world [35,36,37,38]. This fish is generally considered a stenohaline freshwater species; however, it can tolerate low levels of salinity (lower than 8 g kg^−1^) [37] and sometimes pass through brackish-water areas [39,40]. The 96 h half-lethal concentration of salinity to juvenile grass carp was found to be 15.1 g kg^−1^, and adult grass carp survived salinities of 10.5 g kg^−1^ for 24 days and 17.5 g kg^−1^ for 5 h [41,42,43]. Previous studies have observed approximately 3- to 4-fold intraspecific variations in the resting and maximum metabolic rates of this fish [44,45,46]. It remains unclear how salinity tolerance capacity is associated with metabolic parameters among individuals of grass carp. In this study, the salinity tolerance capacity of each individual was characterized by the upper salinity tolerance limit of fish. Moreover, the lowest salinity level that starts to change each characteristic variable of this species, in terms of population, was established as the salinity tolerance threshold of that characteristic variable. We predicted that (1) individuals with lower metabolic rates have a higher upper salinity tolerance limit, and (2) increasing salinity (below the isotonic level) would reduce the resting metabolic rate. We also aimed to see how the structure and physiology of the gills change due to changes in the demands of gaseous and ionic exchange as salinity increases.

Accordingly, we designed two experiments. In Experiment I, the metabolic rates of all individuals of grass carp were determined, and then the upper salinity tolerance limit of each individual was determined by a daily increasing salinity treatment to test whether a negative association existed between the metabolic rates and salinity tolerance capacity among individuals. In Experiment II, six groups of grass carp were treated with different salinities (0, 0.375, 0.75, 1.5, 3, and 6 g kg^−1^) for 2 weeks, and their metabolic rates, gill histology, and Na^+^-K^+^-ATPase activities were determined to evaluate salinity-induced changes in metabolic consumption and changes in the structure and physiology of the gills.

## 2. Materials and Methods

### 2.1. Experimental Fish and Holding

Grass carp (5–25 g) were obtained from a local fish farm located in Chongqing, China. All handling and treatment procedures were conducted according to the ethical requirements for animal care of the School of Life Sciences of Southwest University, China (LS-SWU/012/2016), and the requirements of the environment and housing facilities for laboratory animals in China (Gb/T14925-2001). Upon arrival at the laboratory, the fish were reared in aqua tanks with circulating fresh water for two weeks. The water temperature was adjusted to 25 °C using a thermostat (JRB-250, Sunsun CO., Ltd., Zhoushan, China) and regulated by an automatic temperature controller (PY-SM5, Pinyi CO., Ltd., Yuyao, China). The dissolved oxygen was maintained above 7 mg L^−1^ by continuous air pumping. The water was recirculated through a filtration system (HBL803, Sunsun CO., Ltd., Zhoushan, China), and the ammonia N concentration was lower than 0.015 mg L^−1^. The photoperiods were 12 h light: 12 h dark. During acclimation, the fish were fed to satiation once daily at 18:00 with a commercial diet (33.0% protein, 3.0% fat, 10.0% carbohydrate, and 2.7% salt) (Tongwei Company, Chengdu, China).

### 2.2. Experiment I

Experiment I was designed to examine whether individuals with higher metabolic rates have lower salinity tolerance thresholds. Fifty-four grass carp (body mass 10.0 ± 0.3 g) were randomly selected and individually reared in independent acrylic cells in aqua tanks. The size of each acrylic cell was 40 cm × 25 cm × 20 cm, and there was a high density of holes via the wall to ensure water exchange between the cells and aqua tanks. The fish were acclimatized in the cells for two more weeks before the determination of metabolic rates and salinity tolerance thresholds. At the end of acclimation, the experimental fish were fasted for 48 h and weighed to 0.01 g. Then, the resting metabolic rate, ventilation frequency, and maximum metabolic rate of each fish were determined. After measuring the metabolic rates, the fish were placed back in their original cells. Then, the salinity in the water was adjusted in increments of 1.0 g kg^−1^ per day until all the fish died. The survival of fish was checked at 9:00, 15:00, and 21:00 daily, and the ventilation frequency of each fish was counted by visual observation with a submersible camera (H8S, CST CO., Ltd., Shenzhen, China). The ventilation frequency is positively correlated with oxygen uptake by fish [21,22,47] and can be easily determined with noninvasive visual measurement during rearing daily. In this study, the ventilation frequency was used to approximately reflect the daily changes in oxygen uptake by fish during the whole period of salinity increase. Once the first death occurred, the survival status of the fish was checked every hour. The salinity tolerance (g kg^−1^) of an individual was defined as the maximum salinity level which the fish survived and was corrected using the ratio of survival time (h) at the highest salinity to 24 h according to the following formula:upper salinity tolerance limit (g kg^−1^) = the second-highest salinity (g kg^−1^) + 1.0 (g kg^−1^) × survival hours at highest salinity (h)/24 (h).(1)

The salinity levels were obtained by dissolving commercial synthetic sea salt (Yiyu Aquarium Company, Weifang, China) into water reservoirs and adding the solution to the aqua tanks when replacing the water every day. The water salinity was monitored twice daily using a salinity meter (Mootea AR8012, Smart Sensor Company, Dongguan, China). All individuals died at a salinity of 17 g kg^−1^; therefore, the salinity tolerance treatment lasted 18 days (0–17 g kg^−1^). The fish were fed a commercial diet to satiation once daily at 6:00 p.m. Uneaten pellets in tanks were collected and counted to determine the dry feed intake (g). The daily food feeding rate of each fish was calculated according to the following formula:feeding rate (%body mass) = dry feed intake mass (g)/(initial body mass (g) + final body mass (g)) × 2 × 100(2)

The water temperature was maintained at 25 ± 0.2 °C by a thermal regulator, and the dissolved oxygen concentration of the water was aerated to nearly 100% saturation by continuous air pumping. The water temperature and dissolved oxygen were monitored by an oxygen meter (HQ30D, Hach Company, Loveland, CO, USA).

### 2.3. Experiment II

Experiment II was designed to examine how increasing salinity affects the metabolic rate, gill histology, and activities of Na^+^-K^+^-ATPase in the gills, the intestine, and the kidneys. Sixty fish (body mass 15.7 ± 2.4 g) were randomly selected and acclimatized in aqua tanks for two weeks before salinity treatment. At the end of acclimation, the fish were divided into six treatments (*n* = 10 fish in each treatment) and reared in individual cells in six aqua tanks. The size of each tank was 80 cm × 60 cm × 40 cm. The designed water salinity levels of the six treatments were 0, 0.375, 0.75, 1.5, 3, and 6 g kg^−1^. The salinity was increased by 0.375 g kg^−1^ per day until the desired concentration was reached. The fish were held at their respective salinities for two weeks. The water salinity was monitored twice a day using a salinometer (Mootea AR8012, Smart Sensor Company, Dongguan, China). Water was sufficiently aerated using an aerator, and an oximeter (HQ30D, Hach Company, Loveland, CO, USA) was used to monitor the temperature and dissolved oxygen of the water to maintain the desired conditions (temperature: 25 ± 1 °C; oxygen: >7 mg L^−^^1^). Fifty percent of the water was refreshed with water of the same salinity every day. The fish were fed a commercial diet once daily at a rate of 3% of body mass. At the end of the salinity treatment, the experimental fish were fasted for 48 h and weighed to 0.01 g. Then, the resting metabolic rate, ventilation frequency, and maximum metabolic rate of each fish were determined. After measurement of metabolic rates, the fish were placed back in their original tank overnight before tissue sampling. Then, the fish were anesthetized with Ms-222, and the gills, the kidney, and the intestine were dissected. The left gill was used for morphological analysis, and the right gill, kidney, and intestine were stored in liquid nitrogen for enzyme activity determination.

### 2.4. Protocols of Metabolic Rates

The experimental fish were fasted for 48 h and weighed to 0.01 g. Then, the fish were individually transferred into the chambers of a continuous flow respirometer for oxygen consumption rate determination at 25 ± 0.1 °C. The oxygen consumption determination followed the processes described in a previous study [48]. Each fish was placed in a chamber with a size of 0.13 L overnight. Fourteen chambers were used at the same time, and one more chamber that remained empty was set as a control. All chambers were placed in a water bath (110 cm × 70 cm × 30 cm). Prior to measurement, the water flow rate was adjusted to allow the experimental chamber and control chamber to dissolve the oxygen concentration with a difference of 0.5 to 1 mg L^−1^ and to ensure that the dissolved oxygen concentration of the chamber was higher than 70% saturation. The visual field of the fish was shielded by a water bath to avoid visual interference during the experimental operation. The concentration of dissolved oxygen at the outlet of each chamber was obtained by a dissolved oxygen meter (HQ30D, Hach Company, Loveland, CO, USA). The outflow water from the chamber was collected by a volumetric flask, and the time duration was recorded by a stopwatch to calculate the water flow rate (v, L h^−1^) of each chamber. The rate of individual oxygen consumption for each fish (ṀO_2_, mg O_2_ h^−1^) was calculated by the following formula:ṀO_2_ = ΔO_2_ × v(3)
where ΔO_2_ is the difference in the dissolved oxygen concentration (mg O_2_ L^−1^) between the experimental chamber and the control chamber, and v is the water flow rate through the chamber (L h^−1^). The oxygen consumption rate was measured hourly from 08:00 to 19:00, and the ventilation frequency (times min^−1^) of the fish in the respirometer was visually counted with a stopwatch simultaneously. The lowest three values of the oxygen consumption rate were averaged as the individual resting metabolic rate (mg O_2_ h^−1^), and the simultaneous values of ventilation frequency were averaged as the resting ventilation frequency of the fish.

The maximum metabolic rate of fish was measured by the chasing protocol as described by Wang et al. [48]. After the resting metabolic rate measurement, the fish were placed individually into a circular swimming channel and manually chased for approximately 10 min to exhaustion using a hand net. Then, the fish were transferred back to the respirometer chamber, where the flow rate was adjusted to ensure that 95% of the water in the chamber was refreshed within 1 min. The oxygen consumption rate was measured at 0, 1, 2, 3, 4, 5, 6, 7, 8, 9, 10, 20, 40, and 60 min after transferring the fish back to the chamber. The maximal value of oxygen consumption after exhaustion was used as the individual maximum metabolic rate (mg O_2_ h^−1^). The mass-corrected values of the resting metabolic rate and maximum metabolic rate were adopted by using a mass scaling exponent of 0.75 [49].

### 2.5. Na^+^-K^+^-ATPase Activity

The Na^+^-K^+^-ATPase activity was measured using an ultra-trace Na^+^-K^+^-ATPase analysis kit (A0702-2) (Nanjing Jiancheng Institute of Biological Engineering, Nanjing, China). The gill filaments, intestine, and kidney (0.1 g) were placed in 1.5 mL microcentrifuge tubes containing 0.9 mL of physiological saline solution (5.5 mmol L^−1^ glucose, 124 mmol L^−1^ NaCl, 5 mmol L^−1^ KCl, 203.3 mmol L^−1^ MgCl_6_H_2_O, 1.1 mmol L^−1^ CaCl_2_, and 10 mmol L^−1^ NaHCO_3_). The samples were homogenized using a high-speed homogenizer at 2500 rpm in an ice bath for 1 min. A 25,000 rpm refrigerated centrifuge was used to centrifuge the homogenate for 10 min. The supernatant was taken to determine the Na^+^-K^+^-ATPase activity within 24 h. One unit of Na^+^-K^+^-ATPase activity (U) was defined as the amount of inorganic phosphorus (μmol) produced per min and presented as U g^−1^ tissues.

### 2.6. Gill Microscopy

The second arches of the left gill of three fish of each group were randomly selected for microscopy. The gill arches were dissected and fixed in 4% formaldehyde solution. As the gill size was small, the whole second gill arch was embedded in paraffin, sectioned at 5 μm, and stained with hematoxylin and eosin by AMIDA Biotechnology Inc. (Chongqing, China). The tissue structure was imaged using a VERSA pathology image scanner at 400× (Leica Biosystems, Wetzlar, Germany). For the three gill filaments selected in the second gill arch, heights of ten lamellae (LH, μm) and heights of ten interlamellar cell masses (ILCH, μm) were determined using Image-Pro Plus 6.0 software (Media Cybernetics, Rockville, MD, USA). Then, the averaged values of ILCH and LH were used to calculate the ILCH/LH ratio (%). The changes in the ILCH/LH ratio were used to reflect the relative changes in the gill surface area, a characteristic related to exchanges of both gases and ions.

### 2.7. Statistical Analyses

Data were processed using Excel 2010 (Microsoft Corporation, Redmond, WA, USA). Statistical analyses were performed using R [50]. Data visualizations were performed using the ‘ggplot2′ package [51]. For Experiment I, the changes in ventilation frequency and feeding rate with increasing salinity were analyzed using one-way ANOVA with the LSD test. The change in survival with salinity was described using logit regression of the ‘glm’ function. The survival and death of each individual were quantified to be 1 and 0, respectively. Ordinary least square regression (‘lm’) was performed on the relationships among variables. For Experiment II, one-way ANOVA with the LSD test was performed to analyze the difference in each variable among salinity levels. The level of significance was set to a *p*-value of 0.05.

## 3. Results

### 3.1. Experiment I

The daily feeding rate of individual fish tended to increase and was maintained at high levels (>2% body mass) as the salinity increased from 0 to 6 g kg^−1^ before dropping sharply as the salinity reached 10 g kg^−1^ (salinity effect: *F* = 96.16, *p* < 0.001). The feeding rate at 1 g kg^−1^ (2.83% body mass) was significantly higher than that at 0 g kg^−1^ (2.25% body mass) (Figure 1). Feeding activities ceased when the salinity exceeded 11 g kg^−1^.

The ventilation frequency of individual fish tended to decrease as salinity increased from 0 (26.4 times min^−1^) to 12 g kg^−1^ (10.6 times min^−1^) and then increased as salinity increased to 14 g kg^−1^ (43.7 times min^−1^) (salinity effect: *F* = 85.45, *p* < 0.001; Figure 2). The decreases in ventilation frequency started to be significant at a salinity of 3 g kg^−1^. The ventilation frequency at a salinity of 14 g kg^−1^ was significantly higher than that at any other concentration.

The upper salinity tolerance limit ranged from 13.6 to 16.1 g kg^−1^ among individuals. All fish survived as salinity increased from 0 to 13 g kg^−1^. The first dead fish was recorded when salinity reached 14 g kg^−1^. Survival decreased significantly at a salinity of 15 g kg^−1^ compared with that at 0 g kg^−1^. All individuals died when the salinity reached 17 g kg^−1^. The relationship between the survival status and salinity was described by a logit regression: $survival=1-\frac{1}{1+e^{\left(55.48-3.75salinity\right)}}$ (*Z* = −6.85, *p* < 0.001) (Figure 3). A median lethal salinity of 14.8 g kg^−1^ was predicted according to the regression.

The resting metabolic rate ranged from 105.2 to 216.1 mg O_2_ h^−1^ kg^−1^, while the maximum metabolic rate ranged from 405.6 to 1212.6 mg O_2_ h^−1^ kg^−1^ among the individuals. No significant correlations were observed between the upper salinity tolerance limit and both the resting metabolic rate and maximum metabolic rate (Figure 4). However, the upper salinity tolerance limit was significantly correlated with the ventilation frequency at salinities of 13 g kg^−1^ (*r*^2^ = 0.155, *p* = 0.0032) and 14 g kg^−1^ (*r*^2^ = 0.253, *p* = 0.003) and the initial body mass of grass carp (*r*^2^ = 0.092, *p* = 0.026) (Figure 5).

### 3.2. Experiment II

The resting metabolic rate increased from 47.2 mg O_2_ h^−1^ kg^−1^ to its peak value of 77.8 mg O_2_ h^−1^ kg^−1^ as the salinity increased from 0 to 0.75 g kg^−1^ and then tended to decrease as the salinity increased further (salinity effect: *F* = 8.121, *p* < 0.001, Figure 6). The maximum metabolic rate was not affected by salinity. The resting ventilation frequency changed similarly to the resting metabolic rate as salinity increased (*F* = 14.90, *p* < 0.001).

The interlamellar cell masses enlarged and filled the interlamellar spaces as salinity increased to 0.375–1.5 g kg^−1^ (Figure 7). As the salinity increased further, the interlamellar cell masses atrophied, and more lamellae protruded. The ILCH/LH ratio increased from 30.78% at a salinity of 0 g kg^−1^ to 58.26% at a salinity of 0.375 g kg^−1^ and decreased to 35.11% as the salinity increased to 6 g kg^−1^ (salinity effect: *F* = 9.28, *p* < 0.001, Figure 6).

Salinity induced similar ‘U’-shape changes in the Na^+^-K^+^-ATPase activities of the gills (*F* = 11.26, *p* < 0.001) and digestive tract (*F* = 11.19, *p* < 0.001), but not the kidneys (Figure 8).

## 4. Discussion

The results of Experiment I show that the upper salinity tolerance limit was not correlated with either the resting metabolic rate or maximum metabolic rate among the grass carp individuals. Therefore, this suggests that a lower metabolic rate may not necessarily allow for a better salinity tolerance capacity, which is inconsistent with our prediction. One possible explanation could be that, although a low metabolic rate may reduce the ion exchange between the water and fish body and thus the vulnerability to hypersaline conditions [16,21,22,36], fish with a higher metabolic rate might allocate greater energy to osmoregulation, which may contribute positively to a better salinity tolerance capacity. Therefore, the two opposite mechanisms could be offset from each other. In addition, these metabolic parameters were suggested to be related to fish adaptations to environmental stresses, mostly temperature and hypoxia [17,18,52,53]. The salinity tolerance could have different energetic mechanisms than temperature and oxygen tolerance.

The results also show that grass carp can survive in salinities lower than 14 g kg^−1^ in the frame of daily increasing salinity by 1 g kg^−1^, which is close to the previously reported 96 h half-lethal salinity (15.1 g kg^−1^) of the same species [42], indicating that the gradual incremental design in our study can effectively obtain an upper salinity tolerance limit.

As salinity increased, the grass carp responded by increasing feeding at a salinity threshold of 1 g kg^−1^. The results suggest that a low salinity would not have apparent inhibition on the feeding activities of grass carp. However, feeding activities of most individuals ceased when the salinity exceeded 9 g kg^−1^, which would be harmful to the long-term survival of fish. Therefore, although the grass carp could survive at salinities higher than 9 g kg^−1^ in the short term, longer-term salinity tolerance would not be possible for this fish. These findings can supply reference data for freshwater aquaculture facing increasing water salinity. Similarly, a previous study found that grass carp maintained the feeding rate at salinities lower than 6 g kg^−1^ but reduced the feeding rate at salinities above 9 g kg^−1^ [54]. Indeed, high salinity-reduced food intake has been observed in many studies [55,56,57,58,59,60]. It has been suggested that ion absorption from ingested food is one of the primary pathways of water and salt balance for stenohaline freshwater fish under hypotonic conditions; in contrast, reduced food ingestion is important for relieving the high osmotic stress under hypertonic conditions [42,54], which could explain the reduced feeding rate of grass carp under the high-salinity conditions in our study.

The ventilation frequency of the grass carp started to decrease at a salinity threshold of 3 g kg^−1^. As the salinity increased above 12 g kg^−1^, the ventilation frequency increased acutely, reflecting a stress response [61]. Maintaining osmotic homeostasis with varying salinity is a serious challenge for stenohaline freshwater fish and is an energetically expensive process [24,62]. In hyperosmotic conditions, the resting metabolic rate of fish increases due to the additional costs of maintaining internal homeostasis [29]. Ventilation frequency is one of the parameters controlling the oxygen supply and correlates linearly with the metabolic rate of fish [16,21,23]. The changes in ventilation frequency along with the daily increase in salinity could roughly represent the changes in the metabolic rate of the grass carp. In addition, we found a positive correlation among individuals between salinity tolerance and ventilation frequency at the last two salinity levels (13 and 14 g kg^−1^). This finding suggests that fish with greater regulation of ventilation frequency can be more tolerant to salinity stress. The results also show a positive correlation between initial body mass and the salinity tolerance threshold, suggesting that fish with larger body sizes show better tolerance to salinity.

The metabolic costs of osmoregulation of fish range 10–30% of the resting metabolic rate among species [24,27,63]. Therefore, the response of the resting metabolic rate to increasing salinity may have species-specific characteristics. The results of Experiment II show that the resting metabolic rate of grass carp increased to its maximum value and then decreased to its lowest value with increasing salinity. Similar changes occurred in ventilation frequency with increasing salinity. This can be explained by the fact that respiratory gas exchange and energy consumption tend to decrease with a small increase in salinity due to the saved energy cost of maintaining the balance of osmotic and ionic movements between fish and water [24]. A previous study observed that the growth of grass carp at 3 and 6 g kg^−1^ salinity decreased compared to that in fresh water, which was explained as a greater energy cost of osmoregulation of the fish due to salinity [54]. Our results exclude the assumed reason for the high energetic cost for the reduced growth of grass carp under nearly isotonic conditions. However, the resting metabolic rate and ventilation frequency at a salinity of 6 g kg^−1^ were not significantly different from those at a salinity of 0 g kg^−1^, which could be due to the large interindividual variation with increasing salinity. Similarly, no significant reduction in the resting metabolic rate was found in freshwater European perch (*Perca fluviatilis*) under near-isosmotic water conditions, indicating intraspecific differences in response to salinity [13]. The intraspecific difference in response and tolerance to salinity deserves future studies on the genetics and breeding of grass carp.

An interesting finding of our results was that the resting metabolic rate of grass carp was highest under a very low salinity of 0.75 g kg^−1^. This phenomenon could be due to morphological and physiological restructuring in response to increasing salinity because these processes are energy expensive [64]. As a form of regulation of ion flux and oxygen uptake, the size of the interlamellar cell mass in the gills of some fish changes under varying salinities, which has been observed in several freshwater and seawater fish species, e.g., Atlantic killifish (*Fundulus heteroclitus*), pacamã (*Lophiosilurus alexandri*), Nile tilapia (*Oreochromis niloticus*), mangrove killifish (*Kryptolebias marmoratus*), Arctic grayling (*Thymallus arcticus*), rainbow trout (*Oncorhynchus mykiss*), and threespine stickleback (*Gasterosteus aculeatus*) [19,32,33,65,66,67]. In our results, the structure of filaments of the gills changed under slightly increased salinity. The interlamellar cell masses enlarged and filled the interlamellar spaces under salinities ranging from 0.375 to 1.5 g kg^−1^, and the ILCH/LH ratio was the highest at a salinity of 0.375 g kg^−1^, suggesting marked proliferation of the interlamellar cell tissues (reduced functional lamellae surface area) in the filaments. Therefore, the proliferation of interlamellar cell masses could result in a high resting metabolic rate at a small increase in salinity. As the salinity increased further to 6 g kg^−1^, the interlamellar cell masses atrophied, and more lamellae protruded, implying recovery of the exchange surface. Consistently, the Na^+^-K^+^-ATPase activities of the gills showed a ‘U’-shaped change with increasing salinity. Most Na^+^-K^+^-ATPase is expressed in mitochondria-rich cells in the gill epithelium [11]. Our results suggest that changes in the lamellae surface match the changes in Na^+^-K^+^-ATPase activities. In addition, the Na^+^-K^+^-ATPase activities of the intestine also showed the same ‘U’-shaped change pattern as that of the gills. In fresh water, Na^+^-K^+^-ATPase is generally involved in the uptake of Na^+^ and Cl^-^, while in hypertonic water, Na^+^-K^+^-ATPase may play a role in the secretions of excesses of Na^+^ and Cl^-^ [11,15]. The changing patterns of Na^+^-K^+^-ATPase activities in the gills of freshwater fish with increasing salinity remain to be explained. In our study, the highest salinity was 6 g kg^−1^, which was lower than the internal osmotic pressure of this fish (250–300 mosmol kg^−1^, isosmotic to 7.5–9 g kg^−1^ salinity) [54]. However, considering that fish food contains salt (2.7% in our study), salt intake from food may intensify the demand for ion excretion, even under salinities lower than the internal osmotic pressure of fish. The results of Experiment I show that feeding was enhanced as salinity increased daily from 0 to 6 g kg^−1^, suggesting increased salt intake from food with increasing water salinity.

Previous studies indicated that the maximum metabolic rate is positively correlated with the gill surface area of fish [35]. The maximum metabolic rate could be reduced due to a constraint on the gas exchange capacity of the gill epithelium [29,68]. In our study, the changes in the ILCH/LH ratio suggested changed gill surface areas with increasing salinity. However, the maximum metabolic rate showed no marked changes with increasing salinity, which was different from our expectations because acclimatization to salinity is considered to be an osmo-respiratory compromise. Similar results were also reported in European perch [14]. The constant maximum metabolic rate in our results suggests that the gill gas exchange capacity of grass carp may not be changed by salinity, which allows grass carp to maintain a constant maximum aerobic capacity for important behavioral and physiological activities, e.g., swimming and digestion. It has been reported that the heart rate of grass carp remained constant as the salinity was lower than 8 g kg^−1^ [37], which could maintain constant blood flow to support maximum metabolic rate demands. This finding suggests that salinity-induced changes in the gill surface affect ion exchange capacity more than oxygen uptake capacity.

## 5. Conclusions

In conclusion, this study provides information on the behavior and physiology of grass carp exposed to different salinities. The results suggest that salinity affected both ventilation and food ingestion. As the salinity increased, grass carp responded through gradual changes in their feeding rate, ventilation frequency, and survival. There was no significant correlation between the upper salinity tolerance limit and both resting and maximum metabolic rates among fish individuals. Therefore, a lower metabolic rate may not necessarily allow for a better salinity tolerance capacity. The resting metabolic rate of grass carp increased and then decreased with increasing salinity, which could be due to the relative changes in the interlamellar cell masses and protruding lamellae. The maximum metabolic rate of grass carp remained constant with increasing salinity, which suggests that the salinity-induced changes in the gill surface affect ion exchange capacity more than oxygen uptake capacity.

## Figures and Tables

**Figure 1 animals-11-03445-f001:**
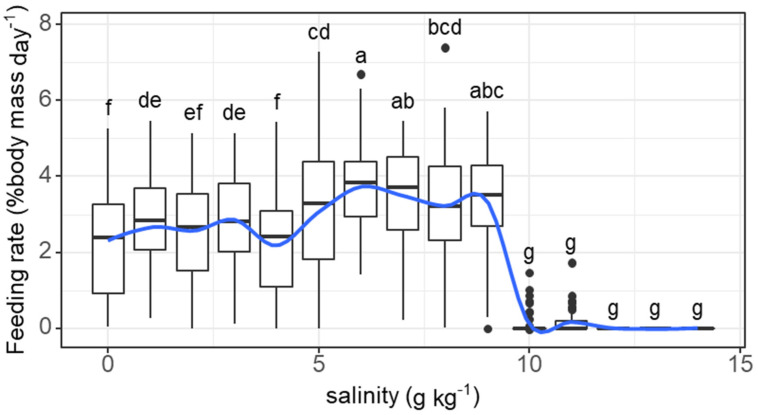
Variations in the feeding rate (% body mass day^−1^) of grass carp with daily increasing salinity (g kg^−1^). The data are expressed as boxplots, with boxes indicating the upper, middle, and lower quartiles, and vertical lines indicating the upper whisker and lower whisker. The blue curve describes the changing tendency using locally weighted scatterplot smoothing (LOWESS). a–g: Boxes without common superscripts indicate significant differences among salinities (*p* < 0.05). Black dot: the outliers.

**Figure 2 animals-11-03445-f002:**
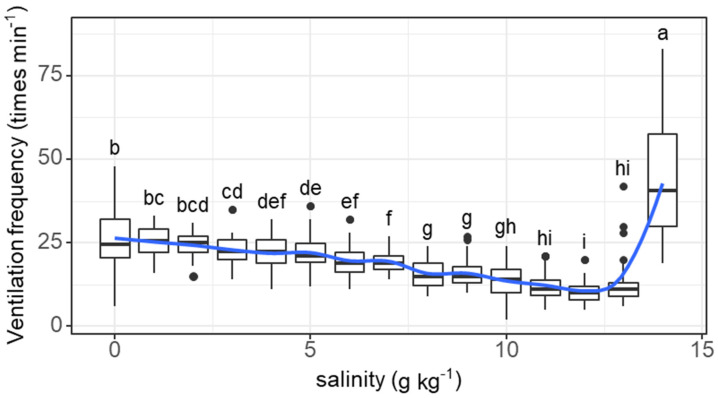
Variations in the ventilation frequency (times min^−1^) of grass carp with daily increasing salinity (g kg^−1^). The data are expressed as boxplots, with boxes indicating the upper, middle, and lower quartiles, and vertical lines indicating the upper whisker and lower whisker. The blue curve describes the changing tendency using locally weighted scatterplot smoothing (LOWESS). a–i: Boxes without common superscripts indicate significant differences among salinities (*p* < 0.05). Black dot: the outliers.

**Figure 3 animals-11-03445-f003:**
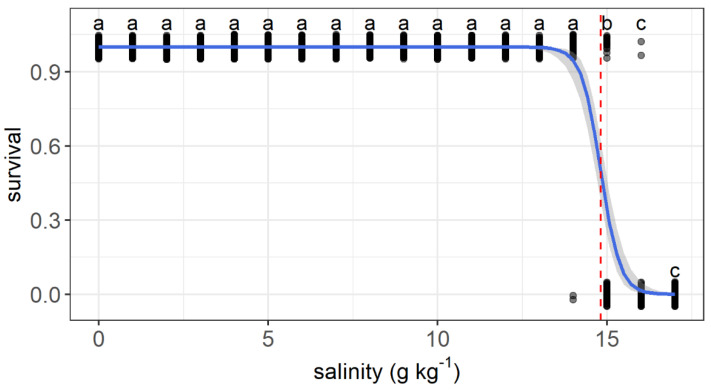
Variations in the survival probability of grass carp with daily increasing salinity (g kg^−1^). The gray dots are used to show the values and are jittered to 0.05 to reduce overlap. The regressions are shown by the blue solid line, and the median lethal salinity is shown by the red dashed lines. a–c: Points without common letter superscripts indicate significant differences among salinities (*p* < 0.05).

**Figure 4 animals-11-03445-f004:**
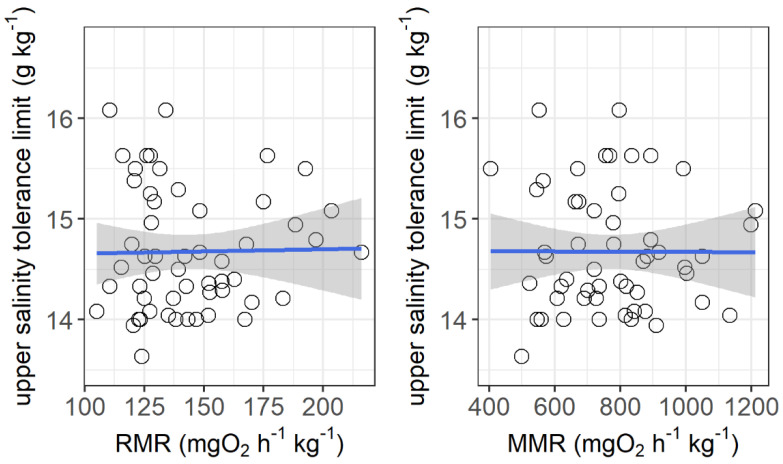
Correlations between upper salinity tolerance limit (g kg^−1^) and resting metabolic rate (RMR, mg O_2_ h^−1^ kg^−1^) and maximum metabolic rate (MMR, mg O_2_ h^−1^ kg^−1^) of grass carp. RMR: *r*^2^ < 0.001, *p* = 0.90; MMR: *r*^2^ < 0.001, *p* = 0.98. The data are shown by the circles and the regressions are shown by the blue line.

**Figure 5 animals-11-03445-f005:**
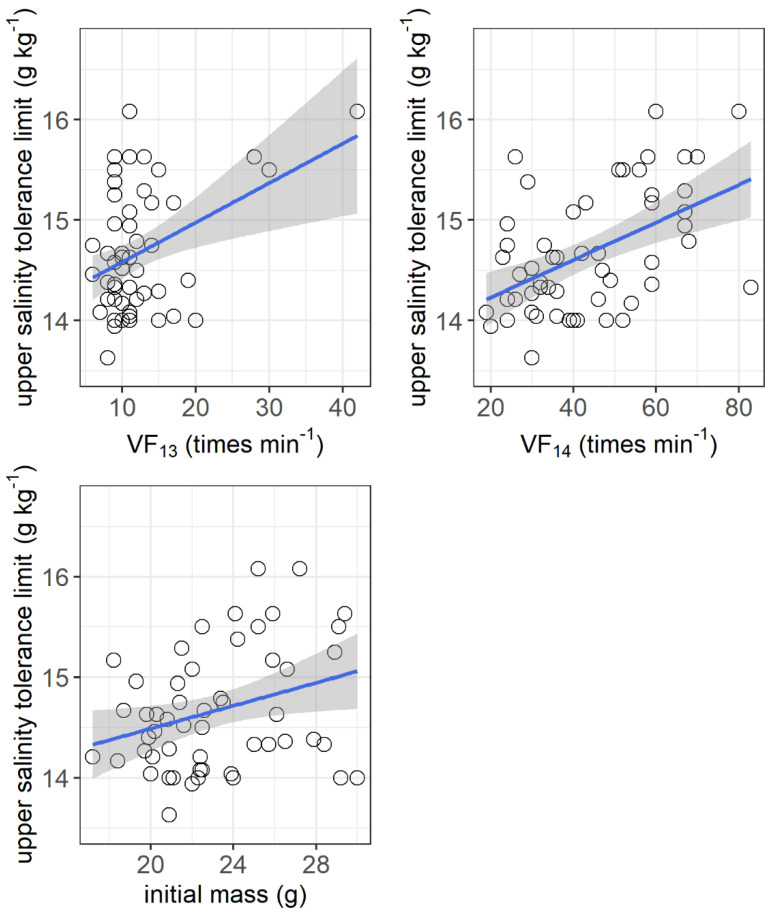
Correlations between upper salinity tolerance limit (g kg^−1^) and ventilation frequency at salinities of 13 g kg^−1^ (VF_13_, times min^−1^) and 14 g kg^−1^ (VF_14_, times min^−1^) and initial body mass (g) of grass carp. VF_13_: *r*^2^ = 0.155, *p* = 0.0032; VF_14_: *r*^2^ = 0.253, *p* = 0.003; initial mass: *r*^2^ = 0.092, *p* = 0.026. The data are shown by the circles and the regressions are shown by the blue line.

**Figure 6 animals-11-03445-f006:**
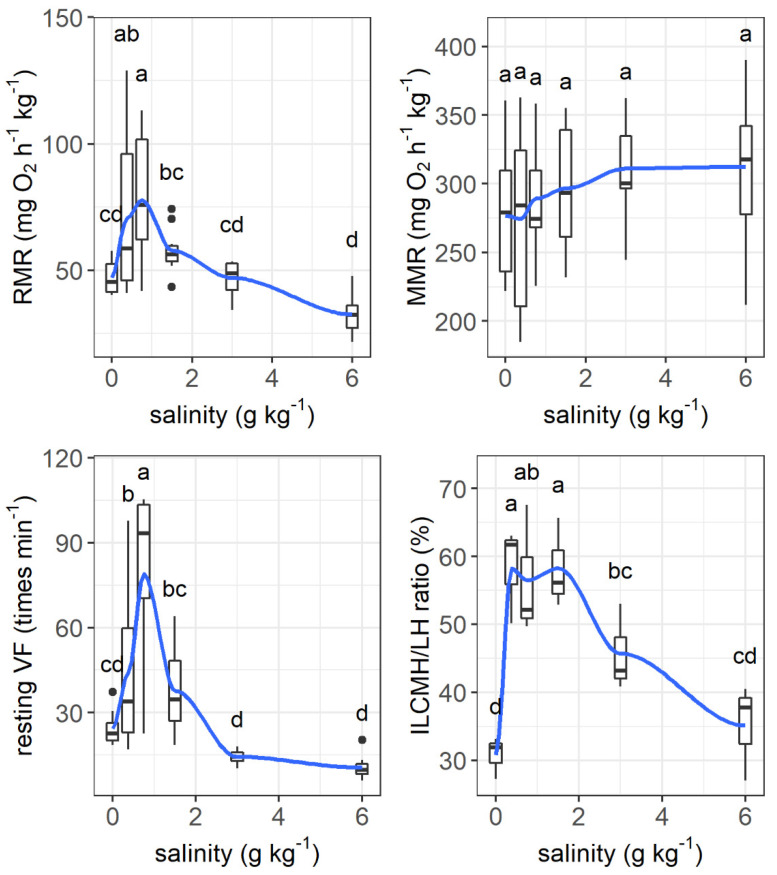
Effects of salinity on the resting metabolic rate (RMR, mg O_2_ h^−1^ kg^−1^), maximum metabolic rate (MMR, mg O_2_ h^−1^ kg^−1^), ventilation frequency (times min^−1^), and the ratio of interlamellar cell mass height to lamellar height (ILCH/LH ratio) of grass carp. The data are expressed as boxplots, with boxes indicating the upper, middle, and lower quartiles, and vertical lines indicating the upper whisker and lower whisker. The blue curve describes the changing tendency using locally weighted scatterplot smoothing (LOWESS). a–d: Boxes without common superscripts indicate significant differences among treatments (*p* < 0.05). Black dot: the outliers.

**Figure 7 animals-11-03445-f007:**
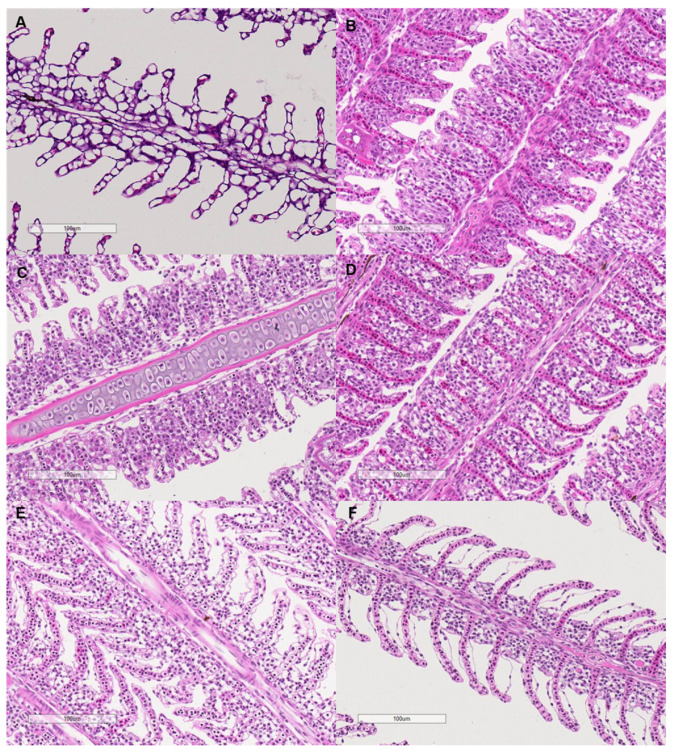
Microscopy of the filament structure of grass carp treated with different salinities (400×). The letters indicate the salinities. (**A**): 0 g kg^−1^; (**B**): 0.375 g kg^−1^; (**C**): 0.75 g kg^−1^; (**D**): 1.5 g kg^−1^; (**E**): 3 g kg^−1^; (**F**): 6 g kg^−1^. The bars represent 100 µm.

**Figure 8 animals-11-03445-f008:**
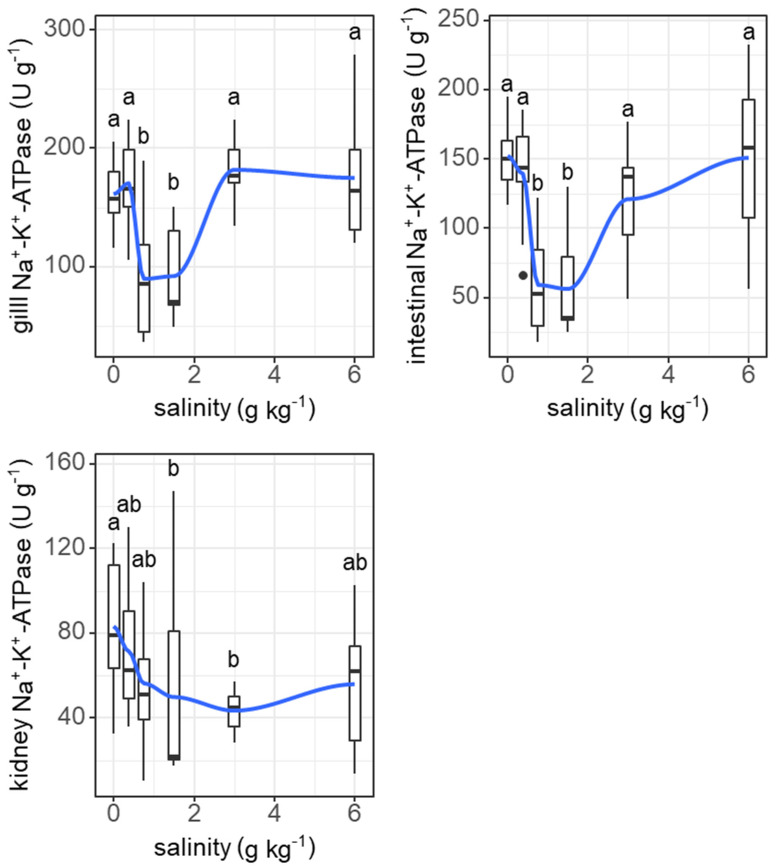
Effects of salinity on Na^+^-K^+^-ATPase activities in gills, intestine, and kidneys of grass carp. The data are expressed as boxplots, with boxes indicating the upper, middle, and lower quartiles, and vertical lines indicating the upper whisker and lower whisker. The blue curve describes the changing tendency using locally weighted scatterplot smoothing (LOWESS). a,b: Boxes without common superscripts indicate significant differences among the treatments (*p* < 0.05). Black dot: the outliers.

## Data Availability

The data presented in this study are openly available in FigShare at doi 10.6084/m9.figshare.14852997.

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
