# Peer review of "Correlation between Metabolic Rate and Salinity Tolerance and Metabolic Response to Salinity in Grass Carp (Ctenopharyngodon idella)"

_animals, 2021, doi:10.3390/ani11123445_

Round 1

Reviewer 1 Report

The authors have improved the quality of the MS and I suggest authors read one more time to fix language errors. Then, it would be ready for the final steps of acceptance.

  • I meant threshold for any parameter you measured and not just repeat this word throughout the MS. I think the authors did not understand what the threshold means. For example, it means in which salinity ventilation frequency will start to change or survival rate. It can be reported for all measured parameters. Please delete the threshold from the text that does not make sense. For example, the correlation between x and salinity tolerance threshold does not make sense. However, you have one formula with this name. Please clear these two points and revise the MS from this point.
  • Line 92, please revise this part.
  • Line 11, you designed this experiment or study designed experiments!!
  • Line 154, g/kg? Is this unit correct?. Are you sure this formula is correct?
  • Line 304, when there is no significant difference no need to report these details. Please update the MS with this change.
  • Figure 4. Is the unit correct? g/kg? Also, please add salinity in the y axis.
  • Figure 4, I think threshold does not make sense for here.
  • Line 361-353, please revise the language.
  • Line 4059, please mention how much is few %
  • Line 419-420, please revise it.
  • Line 457-458, please revise it.
  • Somewhere in the introduction, please explain the difference between “salinity tolerance” and “salinity tolerance threshold” “for parameters.

Best regards

Reviewer 2 Report

General Comments

The manuscript needs work to more clearly explain their focus in the introduction and their conclusions in the discussion.  Further, the authors need to provide rationales for why they measured some of the variables they measured and for their experimental design.

Specific Comments

Introduction

The introduction suffers at the beginning because the authors do not make clear what the salinity range they mean when talk about “salinity change.”  This is important because the effects of salinity change vary depending on the range they mean.  The authors need to make clear from the beginning that they are focusing on the lower salinity range and when they mention change they need to specify the magnitude of the change.

Also, the introduction needs to be simplified to focus on what the authors are studying.  There is a lot of information and not all of it is relevant.

Lines 86 to 87 – It is night clear what they are comparing the higher temperature tolerance to. 

Not to get too picky, but the “hypotheses” that are presented are actually predictions.  Hypotheses focus on the biology of the fish while prediction focus on expected results of experiments.  Predictions stem from hypotheses, but they are not the same.

Line107 – What is “salinity tolerance capacity?”

Methods

It is not clear to me what some of the variables measured have to do with the hypotheses/predictions presented in the introduction.  For instance, the authors predict that fish with lower metabolic rates will have better salinity tolerance.  So what does measurement of ventilation frequency, maximum metabolic rate and aerobic scope have to do with that?  There may be a good rationale for making these measurements, but it was not presented.

Why in the second test did the authors only go up to a salinity of 6 g/kg?  Given the tolerance range it would seem that a maximum salinity around 14 g/kg would have been more informative.  This seems especially important for testing the second prediction.

Results

In general, the presentation of results describes direction, but not magnitude of change.  Can this be corrected?

Lines 273 and 274 – What do “decreased slowly” and “increased rapidly” mean in this context.  Time is not part of the experiment.

Discussion

The hypotheses/predictions presented in the introduction should be the primary focus of the discussion.  The data that test the predictions should be discussed first before discussing other, secondary, results.

I’m not sure about the real significance of the changes in metabolism, ventilation and NKA activities with the small rises in salinity (< 2 g/kg) given the larger salinity range.

I wonder what the feeding results tell us about longer term salinity tolerance of the carp.  They may be about to survive up to 13 g/kg salinities for a week, but if they stop eating at salinities of 9 g/kg then they won’t do well long term.

Line 455 – Can the authors estimate the salt content of the food that fish were fed?

I think the ILCM results are interesting, but I don’t really understand them.  Has anyone ever looked at changes in salinity driving ILCM growth/removal. 

Line 464 – The authors seem to equate gill oxygen permeability with salt permeability.  Is that accurate?

Author Response

This manuscript is a resubmission of an earlier submission. The following is a list of the peer review reports and author responses from that submission.

Round 1

Reviewer 1 Report

General Comments

A look at salinity effects on various aspects of metabolism in the stenohaline FW grass carp.  The study is hampered by what seems to be a lack of clear understanding of the osmoregulatory physiology that underlies the work.  They authors need to sort out these details and revise their manuscript  accordingly.

Specific Comments

Introduction

Line 45 – Provide range of salinities to which they are referring.

Paragraph starting on line 49 – This paragraph needs to be reorganized and expanded to more fully describe the relevant physiology.  In FW fish regulate plasma salt concentrations at levels significantly greater than the surrounding water (hyperregulation).  This leads to diffusive ion losses and osmotic water gain.  Fish deal with this by actively transporting salts from water and excreting water renally.  As salinity increases the difference in salt concentrations becomes less and rates of loss of salts drop and water gain slows.  Somewhere around 13 ppt the water is roughly isosmotic to plasma and any salinity above means the gradients are reversed and a completely different set of mechanisms are required to deal with that (hyporegulation).  Stenohaline fish are stenohaline largely because they do not possess the mechanisms to hypregulate. 

Lines 73 to 76 – This sentence belongs in previous paragraph.

Lines 92 to 94 – I don’t understand this sentence.  In particular, I don’t see what MMR or AS have to do with salinity tolerance.  Salinity tolerance in stenohaline FW fish stems from lack of possession of mechanisms to osmoregulate in waters with salt concentrations higher than plasma levels.  No amount of energy will solve that problem if they do not possess the required mechanisms.

Line 989 – The term hyper-saline usually refers to salinities greater that full strength seawater.

Line 105 – Give range of salinity tolerances you are referring to.

Line 120 – Give range that “large salinity gradients” is referring to.

Line 127 – Plasma ion concentrations were never measured or reported.

Methods

Line 153 – I believe that this is the first time FAS is referred to in the text.  It should be spelled out.

It is not clear in experiment one how long fish spent at each salinity.  The authors state that the salinity was raised 1 ppt per day, but the measurements seemed to take a couple of days to do.

Line 181 – It is not clear how long the fish were held at their respective salinities before measurements were made.  Provide details.

Line 232 – What are AMR and SMR?

Lines 252 to 256 – The sudden mention of interlamellar cell masses begs some explanation of what they are and why they are measured. 

Results

How do the authors determine where the break is in the lines of Fig. 1?  For instance, in the FR line (Fig. 1B) the line looks like it could easily be extended to the next higher salinity. 

Along the same lines why do this sort of analysis instead of simply comparing means with ANOVA?

Fig. 1C – There appear to be 2 points at 15 ppt, a point at 0.0 and another at 1.0.  What is this?

Line 268 – The slope of FR vs. salinity does not appear to be significantly different from zero.

Manu of the r2 values are very low.  That along with the large scatter of the Figs 2 and 4 make any real trends hard to discern.

Paragraph starting on line 320 – Don’t repeat values in text that are presented in figures.  Rather describe any directions and magnitude of differences.

Discussion

Discussion appears to be hampered by lack of understanding of the basic osmoregulatory physiology associated with salinity.  For instance, ventilation rate has little effect on osmosis or diffusion (lines 387 and 388).  Rather, both processes are driven by concentration gradients across the gills.  Ventilation has little effect on that.  Further, the authors refer to “activating ion excretion” at around 12 ppt (line 390), but stenohaline carp probably don’t have any such mechanism.  Rather a more likely explanation is that the rise in VF is stress induced due to possible rising internal salt levels.  By the way, in the introduction the authors refer to measurements of plasma ion levels, but never present any such data.  They would be informative here.

Line 410 – I don’t really understand the utility of estimating “salinity tolerance” in the context of this study.  Given the underlying physiology it could be easily predicted that a stenohaline fish would have a rise in mortality around 14 to 15 ppt because of the switch in the directions of the concentration gradient between plasma and water and the problems that pose for the fish.  They fish do not possess the mechanisms to hyporegulate so once the gradient is reversed they are doomed.

Line 462 – The authors suggest that the rise in NKA activity at 6 ppt is due to “an increase in ion secretion,” yet the fish are still hypertonic to the surrounding water and there is no need for ion secretion at that salinity. 

Line 507 – The “U-shaped change pattern” refers to changes in activity over the whole range of salinities, from 0 to 35 ppt.  The reason it is U-shaped is that in salinities near 13 to 14 ppt the plasma is virtually isosmotic to the surrounding water and active transport in either direction is not needed.  At lower or higher salinities the gradient grows so active transport in one direction or another is required.

Reviewer 2 Report

The authors investigated the correlation between metabolic rate and salinity tolerance and metabolic response to salinity in grass carp. Some major issues significantly compromised the quality of this MS.

Major comments:

  • The way of reporting plots is so confusing and is hard to understand. Generally, readers should be able to understand the data from plots without reading the MS. Therefore, a deep revision is required in plots which I suggested some points in the results section.
  • The manuscript needs to be edited by a native English speaker to improve the language of the MS and fix errors. Some sentences grammatically are correct but hard to understand.
  • Many parts are repetitive, and the authors should revise this MS in terms of brevity.
  • The way of reporting results is so confusing and is not clear. The authors used a bit of complicated statistical analysis and were not able to translate it to something meaning full in terms of biology. I suggest authors change the statistical analysis or rewrite the results to make it clear. The threshold of salinity that impairs fish performance is required, which is missed in this study.
  • Too much abbreviation has caused understanding the results harder. “The RMR was positively correlated with the MMR but negatively correlated with the FAS” we can see this style of writing a lot in this MS. I suggest authors using some of the abbreviations in the complete forms throughout the MS.
  • I am not sure about the accuracy of correlation results as some parameters with r2 of 0.09 and 0.15 are significant. Please double-check the results.
  • Please avoid this style of writing in results “The RMR was significantly affected by salinity”. How was affected? Please write the results clear and direct. Most of the results section is like that and needs to be massively revised.

However, I have touched on some more points that can contribute to the improvement of this MS.

Minor comments

Abstract

  • Line 12-14, please revise this part as you repeated a point twice.
  • Line 14, delete this part.
  • Line 17, delete “among individuals of the grass carp.”
  • Line 20, delete “The results also suggest that”.
  • Line 22, this is not a new concept; delete this part.
  • Line 23, several articles are available on this topic; please revise this part.
  • Line 24, Please provide details about which salinity was considered in each experiment and how long.
  • I suggest the authors gaining some basic knowledge about the metabolic rate and other measured metabolic parameters. Some of these concepts here and in other parts of the MS is not true and should be revised.
  • Please reorder the keywords alphabetically and capitalize each word.
  • Please write the abstract more numerically about the results. You can do it by adding their numbers in parentheses.
  • Please add clearly about the salinity and the experimental period in the abstract.

Introduction:

  • Line 46 and elsewhere, please revise the MS and delete some unnecessary phrases like “it is of interest to understand how.”
  • Line 49-53, too long.
  • What is mean “endangering life cycle processes.”
  • What is internal and external osmotic pressure? Please explain the story here better.
  • “Many fish have developed osmoregulation”. Osmoregulation ??? all fishes have it.
  • Line 59. Delete “Osmoregulation is an energy-expensive process, and”. a lot of words need to be deleted from this MS.
  • Line 60, please shortly explain what is resting metabolic rate.
  • Line 62, delete “such as temperature, dissolved oxygen, and contaminants,”
  • Explain shortly in line 82 what is maximum metabolic rate.
  • Too much abbreviation” please use AS in the complete form. “The difference between the MMR and RMR is called the AS”
  • Line 105-106, please revise it.
  • Line 105, delete “, and it has been farmed in water with salinities higher than those of freshwater”
  • Line 112-114, what is the novelty of your work? This topic was already done in this fish species.
  • Line 114-116, delete this part.
  • Line 118,119, repetitive; please delete it
  • Line 119-121, it is not clear.
  • Line 121-123, it is not true, as numerous studies have been done on this topic.
  • Please review the literature much more carefully and cite more appropriate references. Please check any single reference with reading it instead of citing it without knowing about their concepts.
  • Please update the introduction with recent works as many studies are available from the last two years, which were not included in this section.
  • Line 123, Please mention the novelty of your work in the last paragraph of the introduction.

Material and methods

  • Well-organized section. Clear fellow and all required details were provided.
  • Line 131, fish size
  • Line 139, add the number.
  • Line 148, how many fish per treatment? What are the treatments?
  • Use VAS and FAS in the complete form throughout the MS
  • Line 156, how much is the “desired salinity levels”?
  • Use feeding rate in the complete form.
  • So confusing, it was not clear what is the treatments, how long the fish was exposed? Please explain them here better.
  • Line 117, what is the difference between “effects of salinity on the metabolic rate” in experiments 1 and 2?.
  • Line 181, again, what was the treatment?.
  • Line 183, they are treatments?
  • Line 258, delete this part

Results

  • The way of reporting results or statistical analysis should be updated. Readers want to know whether there was a significant difference in feeding rate, ventilation frequency and others between salinity 0 and 10 or not (For example). From this point, this MS should be deeply revised. Please try to write it clear and straightforward. These kinds of relations and slopes were already predictable; the question is that what is the threshold of salinity for each of these parameters.
  • How R2 of 0.09 is significant? Please double-check the results.
  • Figures: use the same ratio for all tolerance and other parameters. It will help readers to compare the data better.
  • Line 320, please avoid using this style: “The RMR was significantly affected by salinity”. How was affected? Please write the results clear and direct.
  • Please summarize the results, no need to report all details.
  •  

Discussion

  • I did not get through the Discussion section as the previous sections need to be deeply revised.

Best regards